# MOSAIC: Multimodal Object and Semantic Segmentation with Adapter Integration and Contextual Fusion

## Abstract

We introduce MOSAIC, a novel framework for enhancing multimodal RGB-IR object detection and semantic segmentation. MOSAIC utilizes Vision Transformers and introduces modules like the Deformable Feature Sampling, Feature Attention Fusion Block, and Contextual Feature Enhancer. These components dynamically align and integrate RGB-IR features, capturing multi-scale contextual information to enhance object detection and segmentation tasks. Extensive evaluations demonstrate that MOSAIC achieves state-of-the-art results on FLIR, LLVIP, MFNet and VT-series benchmark datasets, significantly improving robustness and accuracy in RGB-IR downstream tasks.

## 1 Introduction

Multimodal imagery analysis enhances object detection by integrating information from multiple modalities, improving robustness and accuracy. While RGB imaging is effective in capturing detailed textures and colors similar to human vision, its performance declines in low-light or adverse conditions such as fog or glare. Infrared (IR) imaging excels in these environments by capturing clear thermal structures but lacks fine texture details. The complementary nature of RGB and IR modalities offers significant advantages in computer vision tasks, making them ideal for fusion in challenging detection scenarios.

While various attempts have been made to fuse RGB and IR features at the feature level to improve object detection performance (Yuan et al., 2022), (Yuan & Wei, 2024), challenges persist, particularly with modality misalignment and inefficiencies in the fusion process. Some approaches address alignment issues through offset estimation and feature integration (Zhang et al., 2025), (Zhang et al., 2019b), or by incorporating scale and angle adjustments for better precision (Yuan et al., 2024b), (Yuan et al., 2022). Cross-attention techniques have been used for spatial correspondence, though they tend to favor global feature fusion. Despite these efforts, there are still significant inefficiencies in fully leveraging the complementary nature of RGB and IR modalities, indicating substantial room for improvement in achieving optimal fusion efficiency and detection performance.

In this paper, we introduce the MOSAIC framework, a novel approach designed to address these challenges through several key contributions:

1. **Deformable Feature Sampling (DFS):** A module to dynamically align RGB and IR features by predicting offsets, which improves robustness.

2. **Feature Attention Fusion Block (FAFB):** A block that integrates RGB-IR features by enhancing them with channel and spatial attention.

3. **Contextual Feature Enhancer (CFE):** An enhancer to seamlessly inject features into a Vision Transformer (ViT) using a progressive, gated strategy.

These contributions enable MOSAIC to achieve state-of-the-art results on benchmark datasets, demonstrating its effectiveness and robustness in multimodal vision tasks.

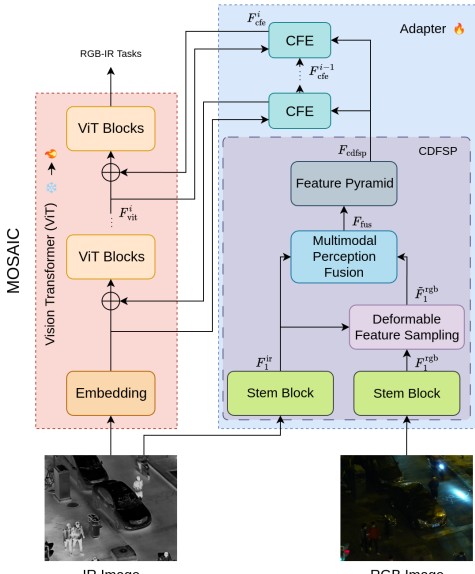

Figure 1: MOSAIC uses ViT to process IR images and integrates multi-scale RGB features via CDFSP. These features are dynamically merged with the ViT outputs using the CFE, enhancing RGB-IR feature representation for downstream applications.

## 2 RELATED WORK

### 2.1 MULTIMODAL OBJECT DETECTION

Early multimodal object detection methods on benchmarks like FLIR (LLC, 2018) and LLVIP (Jia et al., 2021) used two-stream CNNs to fuse features (Zhang et al., 2019a;b). More recently, Transformer-based models such as CFT (Yuan et al., 2024b) and C$^2$Former (Yuan & Wei, 2024) have been introduced to better capture inter- and intra-modality relationships.

### 2.2 MULTIMODAL SEMANTIC SEGMENTATION

In the domain of multimodal semantic segmentation, MFNet (Ha et al., 2017) was one of the early attempts to integrate IR features into an RGB-based framework. Wu et al. (Wu et al., 2022) later introduced CCFFNet, which leverages transformer structures to harness complementary modality features for segmentation tasks. CMX (Zhang et al., 2023) offered a universal cross-modal fusion framework for RGB-IR semantic segmentation, focusing on interactive feature fusion.

### 2.3 MULTIMODAL SALIENT OBJECT DETECTION

CAVER (Pang et al., 2023) re-envisions bi-modal salient object detection through a sequence-to-sequence transformer model, which boosts the interpretability of the system. SwinNet (Liu et al., 2022) utilizes the Swin Transformer to efficiently capture layered features across different modalities, achieving outstanding results. WaveNet (Zhou et al., 2023) employs a distillation technique to inherit extensive knowledge from transformer networks, enhancing its efficiency and compactness.

### 2.4 ADAPTER MECHANISMS

Adapters are lightweight modules inserted into pre-trained transformers for parameter-efficient fine-tuning, a concept first popularized in NLP (Houlsby et al., 2019). This technique has been successfully adapted to vision, with methods like ViT-Adapter (Chen et al., 2023) enhancing ViTs for dense prediction tasks. Recently, UniRGB-IR (Yuan et al., 2024a) demonstrated their effectiveness for RGB-IR tasks by using adapters to inject IR features into RGB-based models. Our work builds on

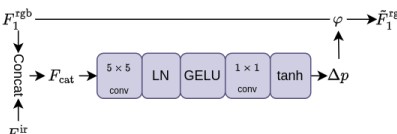

Figure 2: Structure of our Deformable Feature Sampling. It aligns features from different modalities by predicting offsets and sampling the features dynamically, addressing modality miscalibrations effectively.

this foundation, introducing a new adapter-based approach specifically designed for robust RGB-IR feature integration.

## 3 METHOD

### 3.1 OVERALL ARCHITECTURE

As shown in Figure 1, MOSAIC consists of a Vision Transformer (ViT) backbone, a Cross-modal Deformable Feature Sampling Pool (CDFSP), and a Contextual Feature Enhancer (CFE). We use a pre-trained ViT that processes IR images to generate feature tokens. The ViT is frozen during an initial training phase and then fine-tuned. The CDFSP module extracts multi-scale contextual features from both RGB and IR modalities, applying adaptive sampling to the RGB features. These enriched features are then dynamically integrated into the ViT by the CFE module at multiple stages, producing a powerful fused representation for downstream RGB-IR tasks.

### 3.2 CROSS-MODAL DEFORMABLE FEATURE SAMPLING POOL

To enhance feature representations, we introduce the Cross-modal Deformable Feature Sampling Pool (CDFSP), which contains four sequential components. First, a stem block extracts initial features from both modalities. Second, Deformable Feature Sampling aligns the RGB features to the IR features. Third, a Multimodal Perception Fusion module creates and fuses a multi-receptive field representation using various convolution kernels. Finally, a feature pyramid captures multi-scale information, which is particularly beneficial for representing small objects.

**Acquiring Stem Features** We initially use a ResNet (He et al., 2015) stem block to extract features from RGB image ($H \times W \times 3$), resulting in $F_1^{\mathrm{rgb}} \in \mathbb{R}^{H/4 \times W/4 \times C}$. Similarly, we use another stem block to extract IR features $F_1^{\mathrm{ir}} \in \mathbb{R}^{H/4 \times W/4 \times C}$ from IR image.

**Deformable Feature Sampling** As there are modality miscalibrations (Yuan & Wei, 2024) between RGB and Infrared modalities, we use Deformable Feature Sampling to predict the coarse offset between the two modality features and sample the features according to this offset, as illustrated in Figure 2. Initially, we combine the RGB and IR features and use a depthwise separable deviation network to predict their feature offsets. The depthwise separable deviation network consists of two convolution layers, with GELU activation and layer normalization in-between them. These offsets are then used to dynamically sample coarse-aligned features from the RGB and IR feature maps via bilinear interpolation.

The inputs to Deformable Feature Sampling network are $F_1^{\mathrm{rgb}}$ and $F_1^{\mathrm{ir}}$. We concatenate $F_1^{\mathrm{rgb}}$ and $F_1^{\mathrm{ir}}$ into $F_{\mathrm{cat}} \in \mathbb{R}^{H/4 \times W/4 \times 2C}$. It is then passed to the depthwise separable deviation network $f_d$. The first depthwise convolution layer applies a $5 \times 5$ convolution independently to each channel group. The output of the first convolution layer $F_{\mathrm{depth}} \in \mathbb{R}^{H/4 \times W/4 \times C}$ is then layer normalized and passed into GELU activation function. Finally, a second pointwise convolution layer reduces the channel dimension to 2. The output of the second convolution is the offsets $\Delta p \in \mathbb{R}^{H/4 \times W/4 \times 2}$ between the two modalities. We then follow (Yuan & Wei, 2024) to scale the range of $\Delta p$ using the tanh activation function. Thus, the RGB-IR feature offsets are predicted as

$$\Delta p = 2 \cdot \tanh(f_d(\mathrm{Concat}(F_1^{\mathrm{rgb}}, F_1^{\mathrm{ir}})))$$

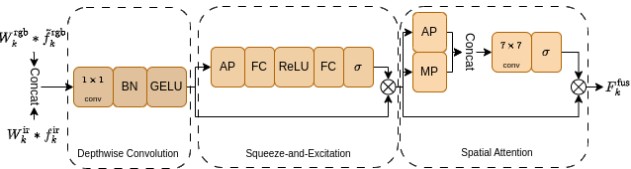

Figure 3: Structure of our Feature Attention Fusion Block (FAFB). It is responsible for effectively integrating and enhancing features from different modalities by employing both channel-level and spatial-level attention mechanisms.

Subsequently, we generate a grid of reference points $r^{\text{rgb}} \in \mathbb{R}^{H/4 \times H/4 \times 2}$ for the RGB modality. These points are normalized to the range [-1, +1], with (-1, -1) representing the top-left corner and (+1, +1) the bottom-right. The RGB stem feature $F_1^{\text{rgb}}$ is sampled at the predicted locations $(r^{\text{rgb}} + \Delta p)$ using the bilinear interpolation sampling function $\varphi(\cdot; \cdot)$ as follows:

$$\tilde{F}_1^{\text{rgb}} = \varphi(F_1^{\text{rgb}}; r^{\text{rgb}} + \Delta p)$$

The $\tilde{F}_1^{\text{rgb}} \in \mathbb{R}^{H/4 \times W/4 \times C}$ is finally passed to the multiple perception fusion network.

**Multimodal Perception Fusion**    To achieve multi-receptive field perception, $\tilde{F}_1^{\text{rgb}}$ and $F_1^{\text{ir}}$ are each first divided into four equal parts of shape $(\frac{H}{4} \times \frac{W}{4} \times \frac{C}{4})$ using the channel splitting technique (Ma et al., 2018). Each part undergoes convolution operations with different kernel sizes ($3 \times 3, 3 \times 3, 5 \times 5, 7 \times 7$). Each processed feature from the two modalities is then fused using a Feature Attention Fusion Block (FAFB).

The FAFB consists of three modules: a depthwise convolution module, followed by a Squeeze-and-Excitation module and spatial module, as shown in Figure 3. In depthwise convolution module, $1 \times 1$ convolution operation is performed within each channel group. It is then followed by a batch norm layer and GELU activation function. The Squeeze-and-Excitation module is responsible for attention at the channel level, explicitly modeling channel interdependencies. The process begins with global average pooling, followed by two fully connected layers interleaved with ReLU activations and an eventual Sigmoid activation. The result is a set of weights that are applied to the input channels by multiplication, endorsing important features while suppressing less informative ones. In the spatial module, the input tensor's channels are pooled via maximum and average operations, creating a 2-channel output that undergoes convolution with a kernel size of 7. The spatial attention map is then generated by applying a Sigmoid activation. This attention map highlights important spatial regions and is used to modulate the input features multiplicatively. By leveraging spatial attention, the network can improve its ability to locate and emphasize relevant spatial cues in the feature representation.

Finally, the fused parts are concatenated to obtain the RGB-IR contextual features $F_{\text{fus}}$:

$$F_{\text{fus}} = \text{Concat}_{k=1}^{4}(\text{FAFB}(W_k^{\text{rgb}} * f_k^{\text{rgb}}, W_k^{\text{ir}} * f_k^{\text{ir}}))$$

where $F_{\text{fus}} \in \mathbb{R}^{H/4 \times W/4 \times C}$, $\tilde{f}_k^{\text{rgb}}$ and $f_k^{\text{ir}}$ are the $k$-th part of $\tilde{F}_1^{\text{rgb}}$ and $F_1^{\text{ir}}$ features respectively, and $W_k$ represents the convolution with $k$-th kernel size.

**Feature Pyramid**    We follow (Yuan et al., 2024a) to apply a series of three $3 \times 3$ convolutions with a stride of 2 to downsample the feature maps. Each scale's features are then passed through a $1 \times 1$ convolution to project the feature maps to $D$ dimensions. These features are flattened and concatenated into feature tokens $F_{\text{cdfsp}} \in \mathbb{R}^{\left(\frac{HW}{8^2} + \frac{HW}{16^2} + \frac{HW}{32^2}\right) \times D}$, which serve as supplementary features for the ViT model.

### 3.3 CONTEXTUAL FEATURE ENHANCER

To seamlessly integrate contextual multi-scale features into the ViT architecture, we introduce the Contextual Feature Enhancer (CFE) module as illustrated in Figure 4. Given that the sequence lengths of the feature tokens from CDFSP ($\frac{HW}{8^2} + \frac{HW}{16^2} + \frac{HW}{32^2}$) and that of the features from ViT

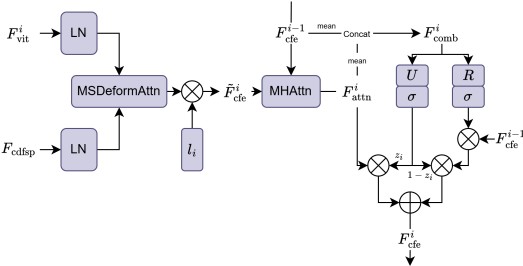

Figure 4: Structure of our Contextual Feature Enhancer (CFE) module. The module integrates supplementary multi-scale features from the Cross-modal Deformable Feature Sampling Pool (CDFSP) into the Vision Transformer (ViT). It ensures progressive feature injection, enhancing feature representation for downstream tasks.

($\frac{HW}{16^2}$) differ, we utilize multi-scale deformable attention (Zhu et al., 2020) to dynamically sample supplementary features. Specifically, we have

$$\tilde{F}_{\text{cfe}}^i = \text{MSDeformAttn}(\text{LN}(F_{\text{vit}}^i), \text{LN}(F_{\text{cdfsp}})) \cdot l_i$$

where $\text{MSDeformAttn}(\cdot)$ represents the multi-scale attention layer, $\text{LN}(\cdot)$ denotes LayerNorm (Ba et al., 2016), and $l_i \in \mathbb{R}^D$ is a trainable parameter for layer scaling.

Inspired by the Gated Recurrent Unit (GRU) architecture, our design adopts a progressive injection strategy to retain and update information from previous output features $F_{\text{cfe}}^{i-1}$. The GRU's gating mechanisms allow it to control the flow of information, making it effective for sequential data processing. For $i = 1$, we initialize $F_{\text{cfe}}^1 = \tilde{F}_{\text{cfe}}^1$. For $i = 2 \ldots N$, we first use multi-head attention layer (Vaswani et al., 2017) to capture dependencies between $\tilde{F}_{\text{cfe}}^i$ and $F_{\text{cfe}}^{i-1}$.

$$F_{\text{attn}}^i = \text{MHAttn}(\tilde{F}_{\text{cfe}}^i, F_{\text{cfe}}^{i-1})$$

Then its mean along the sequence dimension is concatenated with that of $F_{\text{cfe}}^{i-1}$ to obtain $F_{\text{comb}}^i \in \mathbb{R}^{2D}$:

$$F_{\text{comb}}^i = \text{Concat}(\overline{F_{\text{attn}}^i}, \overline{F_{\text{cfe}}^{i-1}})$$

A linear layer $U$ representing an update gate and a linear layer $R$ representing a reset gate are used to obtain the final feature $F_{\text{cfe}}^i$ according to the following equations:

$$z_i = \sigma(U F_{\text{comb}}^i)$$
$$r_i = \sigma(R F_{\text{comb}}^i)$$
$$h_r^i = r_i F_{\text{cfe}}^{i-1}$$
$$F_{\text{cfe}}^i = (1 - z_i) h_r^i + z_i F_{\text{attn}}^i$$

## 4 EXPERIMENTS

To evaluate the effectiveness of our MOSAIC framework, we employ a two-stage training process using the ViT-Base foundation model (Li et al., 2022) pre-trained on the COCO dataset (Lin et al., 2014). In the first stage, the ViT-Base model remains frozen, and only the CDFSP and CFE modules are optimized for 75% of the total training steps. In the second stage, we unfreeze the ViT-Base model to fine-tune it alongside the CDFSP and CFE modules for the remaining training steps. Our method is evaluated and compared against a range of competitive models, including both CNN-based and Transformer-based models.

### 4.1 MULTIMODAL OBJECT DETECTION

**Datasets** For object detection, we use the aligned version of the FLIR dataset (LLC, 2018), which features 4,129 training and 1,013 validation RGB-IR pairs. It covers three classes (person, car, bicycle) at a 512x640 resolution. We also use the LLVIP dataset (Jia et al., 2021) for low-light pedestrian detection, which contains 12,025 training and 3,463 validation pairs at a 1024x1280 resolution.

| Methods | Modality | FLIR | | | LLVIP | | |
|---|---|---|---|---|---|---|---|
| | | mAP | mAP$_{50}$ | mAP$_{75}$ | mAP | mAP$_{50}$ | mAP$_{75}$ |
| SSD | IR | 24.6 | 57.0 | 18.0 | 53.5 | 90.2 | 57.9 |
| RetinaNet | IR | 31.5 | 66.1 | 25.3 | 55.1 | 94.8 | 57.6 |
| Faster R-CNN | IR | 37.6 | 75.8 | 31.6 | 54.5 | 94.6 | 57.6 |
| Cascade R-CNN (ViT-B) | IR | 41.9 | 78.6 | 37.3 | 60.4 | 94.9 | 67.5 |
| DDQ-DETR | IR | 37.1 | 73.9 | 32.2 | 58.6 | 93.9 | 64.6 |
| SSD | RGB | 18.8 | 46.3 | 13.1 | 39.8 | 82.6 | 31.8 |
| RetinaNet | RGB | 21.9 | 51.2 | 15.2 | 42.8 | 88.0 | 34.4 |
| Faster R-CNN | RGB | 27.7 | 62.2 | 21.2 | 45.1 | 87.0 | 41.2 |
| Cascade R-CNN (ViT-B) | RGB | 33.3 | 69.3 | 26.2 | 51.7 | 90.3 | 54.7 |
| DDQ-DETR | RGB | 30.9 | 64.9 | 24.5 | 46.7 | 86.1 | 45.8 |
| GAFF | RGB+IR | 37.4 | 74.6 | 31.3 | 55.8 | 94.0 | 60.2 |
| ProbEn | RGB+IR | 37.9 | 75.5 | 31.8 | 51.5 | 93.4 | 50.2 |
| LGADet | RGB+IR | - | 74.5 | - | - | - | - |
| CSSA | RGB+IR | 41.3 | 79.2 | 37.4 | 59.2 | 94.3 | 66.6 |
| UniRGB-IR | RGB+IR | **44.1** | **81.4** | **40.2** | **63.2** | **96.1** | **72.2** |
| MOSAIC | RGB+IR | **48.9** | **86.8** | **46.7** | **67.7** | **97.6** | **78.7** |

Table 1: Comparison results on the FLIR and LLVIP dataset. The best results are highlighted in **red** and the second-best are highlighted in **blue**. – indicates that the author did not provide the corresponding results.

**Implementation Details**  Our framework is implemented within the MMDetection (Chen et al., 2019) toolbox, utilizing the Cascade R-CNN (Cai & Vasconcelos, 2018) as the base framework for RGB-IR object detection. All experiments are run on NVIDIA GeForce RTX 4090 GPUs. We set the channel dimension $C$ to be 64. The detector is trained with an AdamW optimizer with a learning rate of $1 \times 10^{-4}$, weight decay of 0.1, and layer-wise learning rate decay factor of 0.7 over 12 layers. The first stage of training is run for 36 epochs, with a batch size of 8. The ViT-Base is unfrozen and finetuned for 12 epochs in the second stage of training.

**Metrics**  For quantitative comparison, we use the standard COCO evaluation metric (Lin et al., 2014). Performance is measured by Average Precision (AP), which is based on the Intersection over Union (IoU) between predicted and ground-truth bounding boxes. We report the primary metric, mean Average Precision (mAP), averaged over IoU thresholds from 0.5 to 0.95, as well as AP at specific thresholds of 0.5 (AP$_{50}$) and 0.75 (AP$_{75}$).

**Results**  As summarized in Table 1, our proposed framework significantly outperforms all baseline models on the FLIR and LLVIP datasets. It achieves state-of-the-art mAP scores of 48.9 on FLIR and 67.7 on LLVIP. Notably, our method shows substantial improvements across all metrics compared to the previous best method, UniRGB-IR, with performance gains ranging from 1.6% to 16.2%. These results validate the effectiveness of our framework in leveraging both RGB and IR modalities to achieve superior object detection performance.

## 4.2 MULTIMODAL SEMANTIC SEGMENTATION

**Datasets**  For semantic segmentation, we evaluate on the MFNet dataset (Ha et al., 2017). It provides pixel-level annotations for urban scenes, containing 1,569 day and night images across eight classes. The data is split into training, validation, and test sets with a 2:1:1 ratio.

**Implementation Details**  Similar to the RGB-IR object detection task, we integrate our method within the MMSegmentation (Contributors, 2020) toolbox, utilizing the SETR (Zheng et al., 2021) framework for RGB-IR Semantic Segmentation. The optimizer used is SGD with a learning rate of 0.01, momentum of 0.9, and a weight decay of 0.0005. The learning rate policy follows a polynomial

| Methods | Unlabeled Acc | IoU | Car Acc | IoU | Person Acc | IoU | Bike Acc | IoU | Curve Acc | IoU | Car Stop Acc | IoU | Guardrail Acc | IoU | Color Cone Acc | IoU | Bump Acc | IoU | mAcc | mIoU |
|---|---|---|---|---|---|---|---|---|---|---|---|---|---|---|---|---|---|---|---|---|
| MFNet | 98.7 | 96.9 | 77.2 | 65.9 | 67.0 | 58.9 | 53.9 | 42.9 | 36.2 | 29.9 | 12.5 | 9.9 | 0.1 | 0.0 | 30.3 | 25.2 | 30.0 | 27.7 | 45.1 | 39.7 |
| RTFNet | 99.4 | 98.5 | 93.0 | 87.4 | 79.3 | 70.3 | 76.8 | 62.7 | 60.7 | 45.3 | 38.5 | 29.8 | 0.0 | 0.0 | 45.5 | 29.1 | 74.7 | 55.7 | 63.1 | 53.2 |
| MFFENet | 99.3 | 97.8 | 91.4 | 87.1 | 82.6 | 74.4 | 76.7 | 61.3 | 58.7 | 45.6 | 44.9 | 30.6 | 60.0 | 5.2 | 64.4 | 57.0 | 72.7 | 40.5 | 72.3 | 55.5 |
| EGFNet | 98.7 | 98.0 | 95.8 | 87.6 | 89.0 | 69.8 | 80.6 | 58.8 | 71.5 | 42.8 | 48.7 | 33.8 | 33.6 | 7.0 | 65.3 | 8.3 | 71.1 | 47.1 | 72.7 | 54.8 |
| MTANet | 98.4 | 98.0 | 95.8 | 88.1 | 90.9 | 71.5 | 80.3 | 60.7 | 75.3 | 40.9 | 62.8 | 38.9 | 38.7 | 13.7 | 63.8 | 45.9 | 70.8 | 47.2 | 75.2 | 56.1 |
| MLFNet | - | 97.3 | - | 82.3 | - | 68.1 | - | 67.3 | - | 27.3 | - | 30.4 | - | 15.7 | - | 55.6 | - | 40.1 | - | 53.8 |
| FEANet | 99.3 | 98.2 | 93.3 | 87.8 | 82.7 | 71.1 | 76.7 | 61.1 | 65.5 | 46.5 | 26.6 | 22.1 | 70.8 | 6.6 | 66.6 | 55.4 | 77.3 | 48.9 | 73.2 | 55.3 |
| GMNet | 99.2 | 97.5 | 94.1 | 86.5 | 83.0 | 73.1 | 76.9 | 61.7 | 59.7 | 44.0 | 55.0 | 42.3 | 71.2 | 14.5 | 54.7 | 48.7 | 73.1 | 47.4 | 74.1 | 57.3 |
| CCFFNet | 98.8 | 98.3 | 94.5 | 89.6 | 83.6 | 74.2 | 73.2 | 63.1 | 67.2 | 50.5 | 38.7 | 31.9 | 30.6 | 4.8 | 55.2 | 49.7 | 72.9 | 56.3 | 68.3 | 57.6 |
| CCAFFMNet | 97.4 | 95.2 | 95.0 | 88.6 | 86.1 | 72.9 | 82.1 | 67.1 | 71.2 | 45.9 | 32.1 | 24.8 | 57.1 | 17.8 | 58.0 | 50.1 | 76.2 | 57.8 | 72.8 | 57.8 |
| CMX | - | 98.3 | - | 89.4 | - | 74.8 | - | 64.7 | - | 47.3 | - | 30.1 | - | 8.1 | - | 52.4 | - | 59.4 | - | 58.2 |
| FDCNet | 98.8 | 98.2 | 94.1 | 87.5 | 91.4 | 72.4 | 78.1 | 61.7 | 70.1 | 43.8 | 34.4 | 27.2 | 61.5 | 7.3 | 64.0 | 52.0 | 74.5 | 56.6 | 74.1 | 56.3 |
| ECGFNet | 99.3 | 97.5 | 89.4 | 83.5 | 85.2 | 72.1 | 72.9 | 61.6 | 62.8 | 40.5 | 44.8 | 30.8 | 45.2 | 11.1 | 57.2 | 49.7 | 65.1 | 50.9 | 69.1 | 55.3 |
| LASNet | 97.6 | 97.4 | 94.9 | 84.2 | 81.7 | 67.1 | 82.1 | 56.9 | 70.7 | 41.1 | 56.8 | 39.6 | 59.5 | 18.9 | 58.1 | 48.8 | 77.2 | 40.1 | 75.4 | 54.9 |
| UniRGB-IR | 98.3 | 97.2 | 94.0 | 83.7 | 88.7 | 64.9 | 88.0 | 69.8 | 53.3 | 36.8 | 58.5 | 41.0 | 69.7 | 36.7 | 77.6 | 56.2 | 55.2 | 47.3 | 75.7 | 59.3 |
| RoadFormer+ | - | - | - | - | - | - | - | - | - | - | - | - | - | - | - | - | - | - | - | 62.7 |
| MOSAIC | 98.6 | 97.6 | 90.2 | 82.7 | 90.3 | 65.6 | 81.0 | 66.6 | 71.2 | 54.4 | 69.2 | 54.9 | 84.0 | 69.9 | 76.7 | 53.7 | 70.7 | 56.8 | 81.3 | 66.9 |

Table 2: Comparison results on the MFNet dataset. The best results are highlighted in red and the second-best are highlighted in blue. – indicates that the author did not provide the corresponding results.

decay strategy with a minimum learning rate of 1e-4 with a power of 0.9. The training process runs for a total of 10,000 iterations with batch size of 16.

**Metrics** We assess semantic segmentation performance using mean accuracy (mAcc) and mean intersection over union (mIoU). Mean accuracy reflects the average pixel-wise accuracy across classes, while mIoU measures the average overlap between predicted segments and ground truth.

**Results** The performance of our proposed MOSAIC framework for RGB-IR semantic segmentation on the MFNet dataset is summarized in Table 2. Our method achieves state-of-the-art performance, significantly surpassing previous approaches in both mean accuracy (mAcc) and mean intersection over union (mIoU). Specifically, MOSAIC attains an mAcc of 81.3 and an mIoU of 66.9, which are the highest among all compared methods. Notably, our approach consistently outperforms the previous method, UniRGB-IR, with improvements in mAcc and mIoU of 7.4% and 12.8%, respectively. These results highlight the effectiveness of our approach in integrating RGB and IR modalities to enhance feature representations for semantic segmentation tasks. The substantial gains across various obstacle classes, particularly in challenging categories such as guardrail and car stop, demonstrate the robustness and versatility of the MOSAIC framework.

## 4.3 MULTIMODAL SALIENT OBJECT DETECTION

**Datasets** For salient object detection, we use three standard RGB-IR benchmarks: VT821 (Wang et al., 2018), VT1000 (Tu et al., 2020), and VT5000 (Tu et al., 2023). For the larger VT5000 dataset, we follow the established split of using 2,500 pairs for training and the remainder for testing.

**Metrics** The evaluation of salient object detection performance was based on these metrics: F-measure ($adpF \uparrow$), E-Measure ($adpE \uparrow$), S-Measure ($S \uparrow$), and Mean Absolute Error ($MAE \downarrow$), where $\uparrow$ indicates a higher score is better and $\downarrow$ means a lower score is preferable.

**Settings** We implemented our method using the MMSegmentation library, within the SETR framework. The fine-tuning consisted of 10,000 iterations, starting with a learning rate of 0.01, using the SGD optimizer, and with a batch size of 64. For testing, all images were resized to $224 \times 224$.

**Results** As reported in Table 3, our MOSAIC framework consistently outperforms state-of-the-art methods on the VT5000 datasets across all metrics. Our approach achieves an $S$-measure, $adpE$,

| Model | VT821 | | | | VT1000 | | | | VT5000 | | | |
|---|---|---|---|---|---|---|---|---|---|---|---|---|
| | $S \uparrow$ | $adpE \uparrow$ | $adpF \uparrow$ | $MAE \downarrow$ | $S \uparrow$ | $adpE \uparrow$ | $adpF \uparrow$ | $MAE \downarrow$ | $S \uparrow$ | $adpE \uparrow$ | $adpF \uparrow$ | $MAE \downarrow$ |
| MMCI* | 0.763 | 0.784 | 0.618 | 0.087 | 0.886 | 0.892 | 0.803 | 0.039 | 0.827 | 0.859 | 0.714 | 0.055 |
| TANet* | 0.818 | 0.852 | 0.717 | 0.052 | 0.902 | 0.912 | 0.838 | 0.030 | 0.847 | 0.883 | 0.754 | 0.047 |
| S2MA* | 0.811 | 0.813 | 0.709 | 0.098 | 0.918 | 0.912 | 0.848 | 0.029 | 0.853 | 0.864 | 0.743 | 0.053 |
| JLDCF* | 0.839 | 0.830 | 0.726 | 0.076 | 0.912 | 0.899 | 0.829 | 0.030 | 0.861 | 0.860 | 0.739 | 0.050 |
| MTMR | 0.725 | 0.815 | 0.662 | 0.109 | 0.706 | 0.836 | 0.715 | 0.119 | 0.680 | 0.795 | 0.595 | 0.114 |
| M3S-NIR | 0.723 | 0.859 | 0.734 | 0.140 | 0.726 | 0.827 | 0.717 | 0.145 | 0.652 | 0.780 | 0.575 | 0.168 |
| SGDL | 0.765 | 0.847 | 0.731 | 0.085 | 0.787 | 0.856 | 0.764 | 0.090 | 0.750 | 0.824 | 0.672 | 0.089 |
| FMSF | 0.760 | 0.796 | 0.640 | 0.080 | 0.873 | 0.899 | 0.823 | 0.037 | 0.814 | 0.864 | 0.734 | 0.055 |
| ADF | 0.810 | 0.842 | 0.717 | 0.077 | 0.910 | 0.921 | 0.847 | 0.034 | 0.864 | 0.891 | 0.778 | 0.048 |
| LSNet | 0.877 | **0.911** | **0.827** | 0.070 | 0.924 | 0.936 | 0.887 | 0.022 | 0.876 | 0.916 | 0.827 | 0.036 |
| UniTR | 0.873 | 0.892 | **0.827** | **0.033** | 0.929 | 0.941 | 0.885 | 0.019 | 0.883 | 0.926 | 0.839 | 0.032 |
| UniRGB-IR | **0.881** | 0.895 | 0.806 | 0.039 | **0.939** | **0.943** | **0.894** | **0.018** | **0.906** | **0.935** | **0.849** | **0.027** |
| MOSAIC | 0.878 | 0.904 | 0.822 | 0.034 | **0.941** | 0.940 | **0.889** | **0.018** | **0.916** | **0.937** | **0.852** | **0.025** |

Table 3: RGB-IR salient object detection on VT821, VT1000 and VT5000 datasets. * represents RGB-D SOD transformed into RGB-T SOD. The best results are highlighted in red and the second-best are highlighted in blue.

| Dataset | Method | mAP | $mAP_{50}$ | $mAP_{75}$ |
|---|---|---|---|---|
| FLIR | w/o DFS | 47.1 | 84.9 | 44.8 |
| | w/ DFS | **48.9** | **86.8** | **46.7** |
| LLVIP | w/o DFS | 66.1 | 96.8 | 76.3 |
| | w/ DFS | **67.7** | **97.6** | **78.7** |

Table 4: Ablation study on the effectiveness of Deformable Feature Sampling (DFS) on the FLIR and LLVIP datasets.

| Dataset | Method | mAP | $mAP_{50}$ | $mAP_{75}$ |
|---|---|---|---|---|
| FLIR | DC | 45.0 | 82.7 | 42.5 |
| | DC + SE | 47.2 | 84.9 | 44.4 |
| | Full FAFB | **48.9** | **86.8** | **46.7** |
| LLVIP | DC | 63.5 | 95.6 | 72.2 |
| | DC + SE | 66.5 | 97.0 | 76.8 |
| | Full FAFB | **67.7** | **97.6** | **78.7** |

Table 5: Ablation study on the components of the Feature Attention Fusion Block (FAFB).

*adpF*, and *MAE* of 0.916, 0.937, 0.852, and 0.025 on VT5000, respectively, surpassing all previous methods. Additionally, we achieve results close to the state-of-the-art on the VT821 and VT1000 datasets, further demonstrating the effectiveness of our framework in generating saliency maps that closely align with the ground truths.

## 4.4 ABLATION STUDIES

### 4.4.1 DEFORMABLE FEATURE SAMPLING

We conduct an ablation study to evaluate the impact of the Deformable Feature Sampling (DFS) module in our CDFSP on the performance of the MOSAIC framework. Table 4 shows the results of our experiments on the FLIR and LLVIP datasets. The inclusion of the DFS module consistently improves performance across all metrics. Specifically, on the FLIR dataset, the DFS module improves the mAP by 3.82%, while on the LLVIP dataset, it increases the mAP by 2.42%. The results clearly demonstrate that the DFS module enhances the model's ability to align and integrate features from different modalities, leading to better object detection performance.

| Dataset | Method | mAP | mAP$_{50}$ | mAP$_{75}$ |
|---|---|---|---|---|
| FLIR | w/o Attention | 45.5 | 82.9 | 43.0 |
| | w/o Reset Gate | 46.6 | 84.5 | 44.8 |
| | Full CFE | **48.9** | **86.8** | **46.7** |
| LLVIP | w/o Attention | 64.6 | 96.2 | 74.2 |
| | w/o Reset Gate | 67.0 | 97.1 | 77.6 |
| | Full CFE | **67.7** | **97.6** | **78.7** |

Table 6: Ablation study on the components of the Contextual Feature Enhancer (CFE) on the FLIR and LLVIP datasets.

### 4.4.2 FEATURE ATTENTION FUSION BLOCK

We perform an ablation study to assess the effects of various components within the Feature Attention Fusion Block (FAFB) on the performance of the MOSAIC framework. Table 5 presents the results of our experiments on the FLIR and LLVIP datasets.

1. Depthwise Convolution (DC) Module Only: Using only the depthwise convolution module provides a baseline performance with an mAP of 45.0% on FLIR and 63.5% on LLVIP.

2. DC + Squeeze-and-Excitation (SE) Module: Incorporating the SE module enhances the model's ability to focus on informative channel features, improving the mAP to 47.2% on FLIR and 66.5% on LLVIP.

3. Full FAFB: Implementing the complete FAFB, which includes the spatial module, results in the best performance, achieving an mAP of 48.9% on FLIR and 67.7% on LLVIP.

The results demonstrate that each component of the FAFB contributes to enhancing feature integration. The full block significantly improves the model's ability to capture and leverage important spatial and channel-level features, leading to better object detection performance across different datasets.

### 4.4.3 CONTEXTUAL FEATURE ENHANCER

We analyze the contributions of different components within the Contextual Feature Enhancer (CFE). Table 6 shows the results of our experiments on the FLIR and LLVIP datasets.

1. Without Multi-head Attention: Removing the multi-head attention results in a noticeable decrease in performance, with an mAP of 45.5% on FLIR and 64.6% on LLVIP. This indicates the crucial role of multi-head attention in capturing dependencies across features.

2. Without Reset Gate: Excluding the reset gate shows a moderate performance drop, achieving an mAP of 46.6% on FLIR and 67.0% on LLVIP. This suggests that the reset gate effectively controls the flow of information and helps in refining feature integration.

3. Full CFE: The complete CFE, with both multi-head attention and reset gate, delivers the best performance, attaining an mAP of 48.9% on FLIR and 67.7% on LLVIP.

These results demonstrate that both components are essential for maximizing the feature integration capabilities of the CFE, leading to superior detection performance.

## 5 CONCLUSION

In this paper, we introduced MOSAIC, a framework that significantly improves multimodal RGB-IR object detection and semantic segmentation. By incorporating a Vision Transformer with novel modules such as the Deformable Feature Sampling, Feature Attention Fusion Block and Contextual Feature Enhancer, MOSAIC effectively integrates and enhances RGB-IR features. Our framework achieved state-of-the-art results on several benchmark datasets. Ablation studies validated the benefits of each proposed modules. Overall, MOSAIC enhances the robustness and adaptability of multimodal vision models, paving the way for future advancements in this field.

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

| Dataset | Input modality | mAP | mAP$_{50}$ | mAP$_{75}$ |
|---------|----------------|-----|------------|------------|
| FLIR | RGB | 44.8 | 83.0 | 41.6 |
| | IR | **48.9** | **86.8** | **46.7** |
| LLVIP | RGB | 64.1 | 96.3 | 73.2 |
| | IR | **67.7** | **97.6** | **78.7** |

Table 7: Different input modality for ViT on FLIR and LLVIP.

| Dataset | Train epochs | | mAP | mAP$_{50}$ | mAP$_{75}$ |
|---------|--------------|---------|-----|------------|------------|
| | Stage 1 | Stage 2 | | | |
| FLIR | 48 | 0 | 46.8 | 83.7 | 44.8 |
| | 36 | 12 | **48.9** | **86.8** | **46.7** |
| LLVIP | 48 | 0 | 65.5 | 97.0 | 76.7 |
| | 36 | 12 | **67.7** | **97.6** | **78.7** |

Table 8: Different training strategies on FLIR and LLVIP.

Xizhou Zhu, Weijie Su, Lewei Lu, Bin Li, Xiaogang Wang, and Jifeng Dai. Deformable detr: Deformable transformers for end-to-end object detection. *arXiv preprint arXiv:2010.04159*, 2020.

# A ADDITIONAL ABLATION STUDIES

## A.1 INPUT MODALITY FOR VIT

Table 7 presents the results of our ablation study on the impact of different input modalities for the Vision Transformer (ViT) foundation model on the FLIR and LLVIP datasets. The performance metrics include mean Average Precision (mAP), mAP at 50% IoU (mAP$_{50}$), and mAP at 75% IoU (mAP$_{75}$). Our findings indicate that using infrared (IR) images as input significantly outperforms using RGB images. Specifically, for the FLIR dataset, the IR modality achieves an mAP of 48.9%, mAP$_{50}$ of 86.8%, and mAP$_{75}$ of 46.7%, compared to 44.8%, 83.0%, and 41.6%, respectively, for the RGB modality. Similar improvements are shown on the LLVIP dataset. This improvement can be attributed to the enhanced ability of IR images to capture critical features under challenging conditions such as low illumination and adverse weather, where RGB images typically struggle. The IR modality's superior performance highlights its robustness and effectiveness in extracting relevant features, making it a more suitable choice for multimodal object detection tasks in the ViT foundation model.

## A.2 TRAINING STRATEGY

Our ablation study on the training strategy in Table 8 demonstrates that a two-stage training process significantly enhances model performance compared to a single-stage training approach where the Vision Transformer (ViT) remains frozen throughout. In the first stage, we freeze the ViT model and only train the Cross-modal Deformable Feature Sampling Pool (CDFSP) and Contextual Feature Enhancer (CFE) modules for 36 epochs on the FLIR dataset. In the second stage, we unfreeze the ViT model and fine-tune it along with the CDFSP and CFE modules for an additional 12 epochs. This progressive training approach allows the specialized modules to stabilize and extract meaningful features before jointly optimizing the entire network, resulting in a notable performance improvement. Specifically, we observe an increase in mean Average Precision (mAP), mAP at 50% IoU (mAP$_{50}$), and mAP at 75% IoU (mAP$_{75}$) compared to the baseline where the ViT is frozen for the entire 100 epochs. The two-stage training strategy on the FLIR dataset yields the best results with an mAP of 48.9%, mAP$_{50}$ of 86.8%, and mAP$_{75}$ of 46.7%, highlighting the effectiveness of fine-tuning the ViT model in collaboration with the CDFSP and CFE modules. Two-stage training on the LLVIP dataset shows a similar improvement.

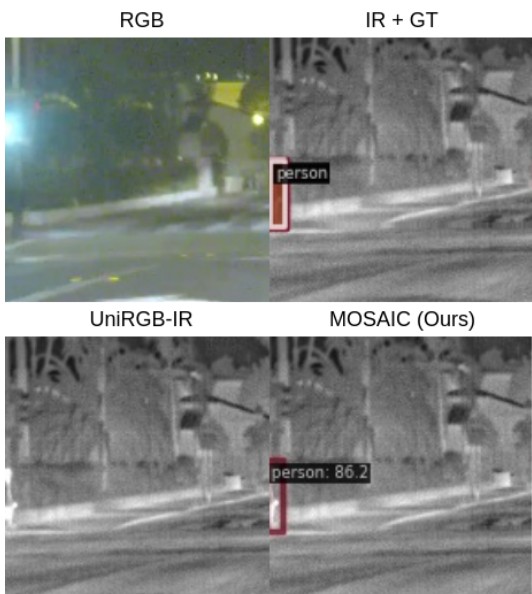

Figure 5: Comparison of object detection results using RGB image, IR image with ground truth (GT), UniRGB-IR, and our proposed MOSAIC framework.

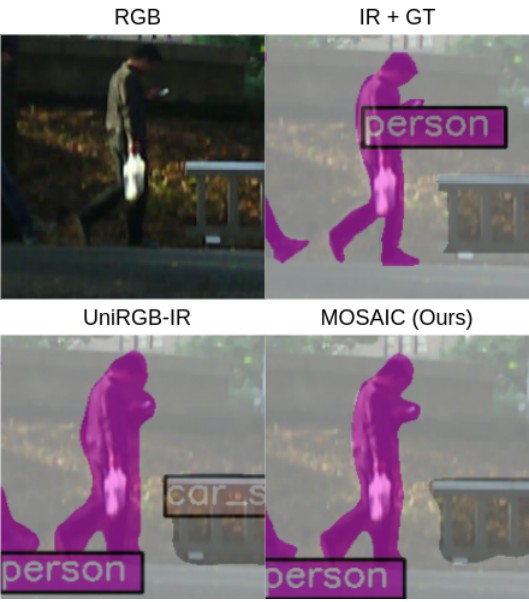

Figure 6: Comparison of semantic segmentation results using RGB image, IR image with ground truth (GT), UniRGB-IR, and our proposed MOSAIC framework.

# B    VISUALIZATION RESULTS

Our visualization results, as shown in Figures 5 and 6, demonstrate the superior performance of the MOSAIC framework compared to the previous best method, UniRGB-IR. In the object detection task, MOSAIC exhibits more confident detections with fewer false positives and more precise bounding box predictions. The bounding boxes are correctly placed with higher confidence scores, and the instances of false detections are significantly reduced. This indicates that MOSAIC can better discern and localize objects in both RGB and IR images. Specifically, in Figure 5, UniRGB-IR failed to detect the person in the IR image due to alignment issues, whereas MOSAIC successfully detected the person, showcasing its improved feature alignment and integration capabilities.

For the semantic segmentation task, MOSAIC provides more accurate and precise segmentations, consistently outperforming UniRGB-IR. The segmentation masks generated by MOSAIC are more aligned with the ground truth, capturing finer details and boundaries of the objects. This results in more accurate identification and delineation of various classes, such as persons, cars, and guardrails. The improved segmentation quality ensures that the contextual understanding of the scenes is enhanced, leading to better performance in downstream applications.

Overall, these results underscore the effectiveness of the MOSAIC framework in integrating RGB and IR features, leading to significant improvements in RGB-IR downstream tasks.