# OpenReview forum: "MOSAIC: Multimodal Object and Semantic Segmentation with Adapter Integration and Contextual Fusion"
_ICLR.cc/2026/Conference — Submitted to ICLR 2026_

### Official Review · Reviewer_h8yJ · 2025-10-28

**Soundness:** 3
**Presentation:** 2
**Contribution:** 1
**Rating:** 2
**Confidence:** 3

**Summary:**

This paper proposes MOSAIC, a multimodal framework for RGB–IR object detection, semantic segmentation, and salient object detection. Cross-modal Deformable Feature Sampling Pool (CDFSP) for RGB–IR feature alignment, Feature Attention Fusion Block (FAFB) for spatial and channel fusion, Contextual Feature Enhancer (CFE) for multi-scale feature injection into a Vision Transformer backbone. Experiments on multiple datasets (FLIR, LLVIP, MFNet, VT5000) show consistent quantitative improvements over prior works such as UniRGB-IR.

**Strengths:**

+ Experiments are conducted on several widely used RGB–IR datasets, showing improvements in multiple tasks.
+ The paper is clearly structured with visual diagrams and ablation studies for the proposed components.
+ The modular decomposition (DFS, FAFB, CFE) is conceptually understandable and potentially extendable.

**Weaknesses:**

- Most components (deformable sampling, SE/spatial attention, adapter injection) are adaptations of known ideas. The paper presents a well-engineered combination but not a fundamentally new mechanism. The “adapter integration” claim is weak since adapters are not deeply analyzed or compared with prior adapter-based fusion models (e.g., UniRGB-IR, ViT-Adapter).
- The CFE module mimics GRU-like gating and multi-head attention without clear motivation or comparison to simpler fusion layers. It is unclear whether performance gains come from architectural complexity or true multimodal synergy.
- The system introduces multiple heavy modules and a two-stage training process. Yet, improvements over UniRGB-IR are small (1–2% mAP / mIoU). There is no runtime or efficiency analysis, making it unclear whether the added complexity is justified.
- The paper claims deformable sampling “aligns RGB–IR features dynamically.”
Can the authors show any visualization or quantitative metric of alignment improvement (e.g., pixel offset statistics or correlation maps)?
- The “contextual fusion” concept is purely empirical; there is no analysis of how deformable offsets, attention, and gating interact. The lack of interpretability or visualization of feature correspondence makes the mechanism opaque.

**Questions:**

Please refer to the Weaknesses.

---

### Official Review · Reviewer_k1LU · 2025-10-28

**Soundness:** 3
**Presentation:** 3
**Contribution:** 2
**Rating:** 4
**Confidence:** 4

**Summary:**

This paper proposes the MOSAIC framework for enhancing RGB-IR multimodal object detection and semantic segmentation tasks. The method is based on the Vision Transformer architecture and introduces three core modules: Deformable Feature Sampling (DFS) for dynamically aligning RGB and IR features, Feature Attention Fusion Block (FAFB) for multi-scale feature fusion, and Context Feature Enhancer (CFE) for injecting fused features into ViT. Experimental results demonstrate that MOSAIC achieves state-of-the-art performance on multiple benchmark datasets including FLIR, LLVIP, and MFNet.

**Strengths:**

1. The paper addresses practical challenges in RGB-IR fusion (modality misalignment, low fusion efficiency, and insufficient utilization of multi-scale information) by proposing specific solutions with clear problem identification. The adoption of a two-stage training strategy (first freezing the ViT to train the adapter modules, then joint fine-tuning) is also a reasonable engineering practice.

2. The experiments are comprehensive with strong results. The paper conducts thorough experimental validation across multiple tasks (object detection, semantic segmentation, and salient object detection) and multiple datasets, demonstrating improvements over previous methods. The ablation studies are also relatively complete, validating the effectiveness of each module.

**Weaknesses:**

1. The paper only demonstrates "what works" but fails to explain "why it works." For example: Why is IR better than RGB as the ViT input (Table 7)? Why is the GRU-inspired gating mechanism necessary? There is a lack of mechanistic explanations and justification for design choices.

2. The paper introduces multiple modules but provides no discussion of the model's computational complexity, inference speed, parameter count, or other efficiency metrics. Compared to baseline methods such as UniRGB-IR, how much additional computational overhead does MOSAIC incur? This is crucial for practical applications.

**Questions:**

1. The three core modules of the paper are direct applications or combinations of existing technologies. Please clarify more explicitly: What is the core technical innovation of this work? Compared to existing work, what are the essential contributions at the methodological or theoretical level?

2. The paper demonstrates that using IR as input for ViT yields better results than RGB, but is this conclusion universally applicable? Does it still hold across different datasets and different scenarios?

3. The paper does not report metrics such as model parameters, FLOPs, inference time, etc. Compared to UniRGB-IR and other baseline methods, what is MOSAIC's computational overhead? This comparison is missing.

---

### Official Review · Reviewer_LdEm · 2025-10-29

**Soundness:** 2
**Presentation:** 2
**Contribution:** 2
**Rating:** 2
**Confidence:** 4

**Summary:**

This paper proposes a framework to enhance multimodal RGB-infrared (IR) vision tasks, including object detection, semantic segmentation, and salient object detection. The main contributions are three introduced key modules on top of ViT to address common challenges of modality misalignment, inefficient feature fusion, and lack of contextual integration in RGB-IR fusion. Experimental results on three multimodal tasks, object detection, semantic segmentation, and salient object detection demonstrate the outperformance of the proposed method.

**Strengths:**

1. This paper is well-organized. The figures and tables are clear and easy to follow the technical details of the proposed key modules.

2. The proposed three modules are technical soundness. Deformable Feature Sampling (DFS) aligns the RGB and IR features by predicting offsets of spatial mis-calibration between modalities. Feature Attention Fusion Block (FAFB) enhances and integrates RGB-IR features via channel and spatial attention. Contextual Feature Enhancer (CFE) utilizes a progressively gated strategy to inject multimodal features into ViT. These modules are validated with extensive ablation studies.

3. The performance improvements on three multimodal RGB-IR tasks, objection detection, semantic segmentation, and salient object detection are consistent according to the reported experimental results.

**Weaknesses:**

**1. Novelty and Motivation**

The overall pipeline of the proposed framework shares a similar motivation and design philosophy with UniRGB-IR [1], as both aim to achieve effective RGB-IR feature fusion. For example, UniRGB-IR also employs a feature pyramid, element-wise and channel-wise multiplication, and a gating mechanism for multimodal feature integration. Although MOSAIC introduces three new modules—Deformable Feature Sampling (DFS), Feature Attention Fusion Block (FAFB), and Contextual Feature Enhancer (CFE)—these components appear to be combinations or extensions of existing techniques such as [2], [3], [4], [5]. As a result, the overall contribution may not present a substantially novel motivation or unique architectural innovation for multimodal RGB-IR tasks.

**2. Fairness of Experimental Comparison**

The experimental setup raises concerns regarding fairness in comparison with UniRGB-IR [1]. Specifically, the proposed MOSAIC framework adopts a two-stage training strategy, where the ViT backbone is frozen in the first stage and later unfrozen for fine-tuning. In contrast, the baseline UniRGB-IR keeps the ViT backbone frozen throughout training. This difference provides MOSAIC with additional optimization flexibility, potentially inflating performance gains unrelated to the proposed modules. Furthermore, it is unclear why fine-tuning the ViT backbone is necessary, given that adapter-based frameworks are generally designed to avoid retraining large backbones.
Another source of inconsistency lies in the input modality: the proposed method uses IR as the input to the ViT backbone, while UniRGB-IR [1] uses RGB as the primary input, which is the more general and standard setup. For a fair comparison, both methods should maintain consistent input modalities and training strategies. As indicated in Tables 7 and 8, performance drops significantly when MOSAIC uses RGB inputs or when ViT fine-tuning is disabled, suggesting that the reported gains may partly stem from these training and input discrepancies rather than the architectural design itself.

**3. Efficiency and Model Complexity**

The proposed CDFSP and CFE modules introduce additional parameters into the adapter structure, which likely impact the computational efficiency and overall model complexity. An analysis of FLOPs, parameter count, or inference latency compared to UniRGB-IR and other baselines would help clarify the practical trade-offs of the proposed approach.

**References**

[1] Yuan, Maoxun, et al. "UniRGB-IR: A Unified Framework for Visible-Infrared Semantic Tasks via Adapter Tuning." arXiv preprint arXiv:2404.17360 (2024).

[2] Dai, Jifeng, et al. "Deformable convolutional networks." Proceedings of the IEEE international conference on computer vision. 2017.

[3] Zhu, Xizhou, et al. "Deformable detr: Deformable transformers for end-to-end object detection." arXiv preprint arXiv:2010.04159 (2020).

[4] Lin, Tsung-Yi, et al. "Feature pyramid networks for object detection." Proceedings of the IEEE conference on computer vision and pattern recognition. 2017.

[5] Dey, Rahul, and Fathi M. Salem. "Gate-variants of gated recurrent unit (GRU) neural networks." 2017 IEEE 60th international midwest symposium on circuits and systems (MWSCAS). IEEE, 2017.

**Questions:**

**1. Clarification of Differences**

Please provide a clearer and more detailed explanation of how the proposed MOSAIC framework differs from the existing UniRGB-IR approach [1].

**2. Additional Comparison with UniRGB-IR**

It would strengthen the empirical validation if you could include more experimental results for UniRGB-IR when using the IR input modality.

**3. Efficiency Analysis**

Please include an efficiency comparison between the proposed method and relevant state-of-the-art baselines. Metrics such as parameter count, FLOPs, inference time, and memory usage would provide valuable insights into the computational cost and practical deployment feasibility of MOSAIC.

**4. Qualitative and Ablation Studies for DFS**

For the Deformable Feature Sampling (DFS) module, additional qualitative analyses (e.g., visualization of feature alignment or sampling offsets) and detailed ablation results would help demonstrate its effectiveness and clarify how it contributes to performance gains.

**Others comments**

Please ensure that all sota methods are clearly cited in the Experiments section. In the current version, it is difficult for readers to identify the corresponding references for some of the compared methods.


[1] Yuan, Maoxun, et al. "UniRGB-IR: A Unified Framework for Visible-Infrared Semantic Tasks via Adapter Tuning." arXiv preprint arXiv:2404.17360 (2024).

---

### Official Review · Reviewer_C5mv · 2025-11-01

**Soundness:** 2
**Presentation:** 2
**Contribution:** 2
**Rating:** 4
**Confidence:** 5

**Summary:**

This paper presents the MOSAIC framework, which targets multimodal object detection and semantic segmentation by combining RGB and infrared (IR) information. The method uses Vision Transformers as the backbone and introduces three key modules: Deformable Feature Sampling (DFS) for dynamic feature alignment, Feature Attention Fusion Block (FAFB) to improve multi-scale fusion through combined channel and spatial attention, and Contextual Feature Enhancer (CFE) for progressive, gated feature injection. The authors conduct extensive experiments and ablation studies, showing that MOSAIC consistently outperforms several baselines and prior state-of-the-art methods on FLIR, LLVIP, MFNet, and VT-series benchmarks.

**Strengths:**

1. The paper introduces a well-motivated, modular approach to RGB-IR fusion, with each component (DFS, FAFB, CFE) described in sufficient technical detail and mapped clearly to its overall architecture (see Figure 1, page 2).
2. The paper is easy to follow.

**Weaknesses:**

1. Some of the methods referenced in the paper do not include corresponding citation details.
2. The paper focuses exclusively on the RGB–IR case, which is a well-motivated choice. However, it is unclear whether the proposed modules and fusion mechanisms would generalize to other modalities, such as RGB–D or RGB–X, or to cases with different alignment characteristics. Including some discussion or empirical evidence on the generalizability of the approach would strengthen the broader impact of the work.
3. Some reporting in Table 2 (page 7) is hard to interpret due to formatting errors (e.g., merged columns/entries like '771.176 .761', inconsistent use of separators)
4. The paper employs a two-stage training strategy, but provides limited justification for hyperparameter choices, strategies to avoid overfitting, or how the validation set is tailored beyond specifying batch size, epochs, and optimizer. There is also little discussion of model selection or tuning practices, such as early stopping criteria or the use of data augmentations. Clarifying these aspects would improve reproducibility and give more confidence in the reported results.

**Questions:**

Please refer to the weaknesses section.

---

### Meta-Review · Area_Chair_Ngqt · 2026-01-07

**Summary:**

The reviewers questioned the novelty of the proposed modules, the fairness of the experimental comparison and the practical value of the approach. The authors did not participate in the discussion, so all concerns remain unresolved. I recommend rejection.

**Reviewer Concerns:**

None of the substantive concerns raised in the reviews were resolved, as the authors did not engage in the discussion or provide additional clarification.

**Reviewer Scores:**

All scores remain unchanged, resulting in two rejections and two marginal recommendations still below the threshold.

---

### Decision · Program_Chairs · 2026-01-26

Reject